# Shallow coastal zones are key mediators in Arctic land-ocean carbon fluxes

F. C. J. van Crimpen [1] ✉, L. Madaj[1], J. M. van Genuchten [2], T. Tesi[3], D. Whalen [4], K. Scharffenberg[5], L. Bröder[6], M. Fritz [7] & J. E. Vonk [1]

Rapid Arctic warming accelerates the erosion of permafrost coasts rich in terrestrial organic carbon (terrOC). Once released into the ocean, terrOC can degrade or get buried in shelf sediments, yet its transport pathways and fate remain poorly understood. We collected permafrost material, sediment and surface water along the Canadian Beaufort Sea coast, fractionating samples by density (cut-off 1.8 g/cm$^3$) and size (38, 63 and 200 µm) before performing geochemical and microscopic analysis. Our results show that ~43% of terrOC is trapped in low-density fractions, mainly as vascular plant debris. Surprisingly, this material is trapped within shallow (0-5 m) waters where waterlogging and large particle size increase its density and settling velocity. Less than 10% is transported to deeper waters (30-55 m), indicating that the shallow coastal zone acts as a trap and biogeochemical reactor. These findings challenge the source-to-sink paradigm and highlight the overlooked and undersampled (< 6% of pan-arctic shelf data) nearshore zone.

The Arctic region is undergoing rapid environmental changes driven by global climate warming, with implications for the carbon cycle and climate feedback mechanisms[1–4]. One of the critical consequences of accelerated warming is the increased release of terrestrial organic carbon (terrOC) from thawing permafrost soils[3,5,6]. Current estimates suggest that permafrost stores ~1300 pg of organic carbon (OC)[7], part of which is vulnerable to mobilization and degradation as permafrost thaws and erodes[8,9].

Erosion of Arctic permafrost coasts releases terrOC and nutrients into the marine system, where it supports and alters ecosystem functioning[10–13]. Once transported into the Arctic Ocean, terrOC can either be remineralized within the water column or buried in marine sediments[14–16]. The fate of this OC determines the eventual contribution to atmospheric $CO_2$ levels and overall climate impact[1,8].

TerrOC transport in marine systems is driven by waves and currents, creating contrasting hydrodynamic conditions that separate and sort eroded material by density, size, and mineral associations[17–20]. Recent studies suggest that a large fraction of terrOC, such as vascular plant material, resides in the low-density fraction on land[18,21,22], which is thought to be particularly vulnerable to degradation as it is not protected by the mineral matrix that can stabilize OC through OC-mineral associations[23,24]. Small and light fractions of this material can remain buoyant over longer distances when it is not ballasted by minerals[25]. Shallow coastal zones are vital for winnowing, breakdown, and sorting of this OC-rich material[21,22,26] and are critical for trapping and holding terrOC, and function as an active filter before the remaining material eventually enters the long-distance sediment transport[18,21,27].

This study examines the shallow coastal zone of the Canadian Beaufort Sea (Fig. 1), specifically the 0–5 m zone, which has been divided into 0–2 and 2–5 m water depth to understand the hydrodynamic pathways of permafrost-derived terrOC. The 0–2 m zone is wave-dominated, leading to resuspension of eroded material into the water column, while the 2–5 m zone is less energetic and shows a more pronounced settling pattern.

Using a transect-based approach, sea surface sediment and surface water samples were collected from five locations along a 2030 km coastal stretch[22], supplemented by sediment trap data from the Mackenzie Delta and surface sediment samples from the Beaufort Sea shelf (30–55 m depth)[28]. To trace the fate of terrOC, we used hydrodynamic fractionation (density cut-off 1.8 g/cm$^3$ combined with wet-sieving at 38, 63 and 200 µm) to separate sediments before further geochemical (OC, total nitrogen (TN), and δ$^{13}$C–OC) and visual analyses with a scanning electron microscope (SEM). The geochemical analysis, together with SEM imaging, lets us further understand the relative contribution of mineral-associated OC vs vascular plant debris.

By doing so, we aim to obtain a quantitative estimate and improve our understanding of the organic matter distribution and its fate among

[1]Department of Earth sciences, Vrije Universiteit Amsterdam, Amsterdam, The Netherlands. [2]Department of Geosciences, Centre for Ice, Cryosphere, Carbon, and Climate (iC3), The Arctic University of Norway, Tromsø, Norway. [3]National Research Council, Institute of Polar Sciences, Bologna, Italy. [4]Geological Survey of Canada, Natural Resources Canada, Dartmouth, NS, Canada. [5]Department of Fisheries and Oceans Canada, Winnipeg, MB, Canada. [6]Department of Earth and Planetary Science, Geological Institute, Swiss Federal Institute of Technology (ETH), Zurich, Switzerland. [7]Permafrost Research Section, Alfred Wegener Institute, Helmholtz Centre for Polar and Marine Research, Potsdam, Germany. ✉e-mail: f.c.j.van.crimpen@vu.nl

different sediment size and density fractions and a qualitative insight into what this material may consist of. Here, we divert from conventional analyses on bulk samples that cannot provide the information to disentangle hydrodynamic sorting from degradation[18,29].

Gaining knowledge about the fate of terrOC at the interface between land and ocean is crucial for evaluating the effects of increased terrOC fluxes from eroding permafrost coastlines and Arctic fluvial systems[30–33]. These processes have in common that they are major suppliers of terrOC into the nearshore zone and will ultimately help characterize the carbon-climate feedback resulting from thawing permafrost.

## Results

We found distinct patterns in the OC distribution among hydrodynamic fractions of sediments, in surface water samples and in comparison with its permafrost parent material along the Canadian Beaufort Sea. Based on mass partitioning among the different fractions from land to shelf, we show that the low-density (LD) material is abundant in the permafrost parent material and decreases towards deeper water depths across the shelf (Table 1). The high-density coarse (HD > 1.8 g/cm$^3$, >63 μm) material is abundant on land and in the shallow zones (0–2 and 2–5 m). Finer material (>1.8 g/cm$^3$, <63 μm) dominates the shelf sediments and water column. Yet in the shallow regions, each site exhibits its own trend in mass partitioning of the fractions, which suggests that they are affected by different hydrodynamic regimes that can cause either bypassing or trapping of fine material.

Across all sites, the suspended material within the water column is rich in very fine material (HD < 38 μm), except for McKinley Bay, where the HD coarse fraction is dominant, suggesting a high-energy system. The sediment trap in Shallow Bay (Fig. 1) that collected material from the Mackenzie River shows a similar pattern to the shelf sediments, with most of the weight residing in the HD < 38 μm fraction and a decreasing contribution of the fractions with coarser grain size.

Regarding OC contents (%OC), the LD fraction exhibits the highest values in all zones (from land to shelf) and across all sites. OC content (%) decreases with increasing density (HD coarse fractions) and grain size. Similar trends were found by Jong et al. [21] for a transect off Herschel Island, close to the Yukon Coast, with high %OC in the LD fraction and lower values for the HD < 38 μm and HD 63–200 μm fractions. This supports the assumption of the "mineral nature" of the fine fraction, where most of the carbon is bound to the surfaces of inorganic particles[24]. It is worth noticing that in our study the OC% in the HD > 63 μm fraction is highly variable, particularly in the 2–5 m zone, and regularly exceeds 1.0% and sometimes even 10% (Table 1 and supplementary information Fig. 4). These relatively high contributions of OC in traditionally OC-poor mineral fractions suggests that these fractions also hold vascular plant debris, which is supported by SEM images that show pieces of vegetation debris, roots, and leaf fragments. These %OC patterns (Table 1), however, vary between sites depending on the coastal type and local environmental conditions[22].

By combining the weight distributions with the OC content of the specific fractions, we can evaluate the relative importance of each fraction for the overall bulk OC (i.e., OC-partitioning). This shows that the median contribution of LD to the total OC-partitioning decreases from land towards the shelf with 43 ± 22% (median ± IQR) in parent material to 21 ± 29% and 29 ± 22% in the 0–2 and 2–5 m nearshore zones, respectively, and 11 ± 6% on the shelf (Fig. 2). Surprisingly, the coarser HD > 63–200 μm fraction contributes up to 32 ± 40% of the OC-partitioning in the 0–2 m zone and 11 ± 17% in the 2–5 m zone (Table 1 and Fig. 2). The HD > 200 μm fraction contributes 50 ± 35% of the total OC in the 0–2 m zone, and 11 ± 56% in the 2–5 m zone.

Furthermore, δ$^{13}$C–OC values within coarser fractions in the 0–2 m zone are consistently more depleted (HD fractions ranging from −26.5‰ to −28.3‰ VPDB (Table 1)) than the LD or HD < 38 μm fractions (−26.2‰ and 26.5‰), suggesting that the HD coarse fractions contain fresh terrestrial material (Supplementary information Fig. 2). These δ$^{13}$C–OC patterns are still present but less pronounced in the HD coarse fractions in the 2–5 m

zone (−27.1‰ to 27.9‰) but absent on the shelf (−25.9‰ to −22.0‰) (Supplementary information Fig. 2).

On the shelf, the contribution of OC in the coarse HD fractions has decreased to 0.91 ± 1.4% and 0.51 ± 0.8% for the HD > 63 μm and the HD > 200 μm fractions, respectively (Table 1). For the suspended material in the water column, the HD < 38 μm fraction is the main contributor of OC for both the 0–2 m and the 2–5 m zone (with 76 ± 44% and 99 ± 36%, respectively).

Our results also reveal a high heterogeneity among different fractions for the different sites (Fig. 2). We infer that the transport in the 0–2 and 2–5 m zones reflects different wave regimes that vary depending on the coastal setting, making them act as either a bypass or an accumulation zone[34]. Examples of accumulation zones for HD coarse sediment include Toker Point and McKinley Bay, where water depths increase rapidly (~1.5 km to 5 m water depth), likely promoting wave-driven sediment transport for the finer fractions and deposition of predominantly coarse material (Figs. 2 and 3). This is supported by the presence of relatively coarse material in suspension in the water column (HD 63–200 μm and HD > 200 μm making up between 4.86 ± 53% and 2.13 ± 25% of total weight) in both Toker Point and McKinley Bay (Supplementary information Fig. 1). By contrast, at shallower locations such as Tent, Pelly, and Tuktoyaktuk Islands, wave energy is quickly reduced further offshore by the relatively flat bathymetry (between 4 and 36 km to 5 m water depth) (Fig. 1) so that HD < 38 μm fine sediments (88.3% and 99.4% of total weight in the HD < 38 μm fraction) are retained within the 0–5 m zone. Despite these large differences in coastal dynamics across different sites, the material deposited on the shelf remains strikingly similar and is dominated by fine material (Fig. 3), which is rich in mineral-associated OC (Figs. 2, 3). This highlights the critical and complex role of the first few meters offshore in determining the fate of terrestrial permafrost OC (Fig. 4).

## Discussion

Traditionally, studies on the fate of terrOC in the Arctic Ocean have focused on the "source-to-sink" paradigm (Fig. 4), comparing the composition of terrestrial material at the source with its final deposition in continental margin sediments[19,35–37] A key assumption underlying this approach is that the transformation of terrOC along the land-ocean continuum is primarily driven by its reactivity[38] so that shelf sediments offer a direct archive to assess how terrOC changes (degrades) upon its release into the Arctic Ocean and along the transport path. Furthermore, it is often assumed that most OC resides within the mobile fine-grained fractions (HD < 63 μm) as mineral-associated carbon. With the addition of surface area analysis, degradation of this carbon can be precisely compared over (river-dominated) spatial transects[39–41].

Our findings from the Canadian Beaufort Sea region challenge these assumptions (Fig. 5). We demonstrate that a substantial fraction of OC (about 43 ± 22%) coming from both coastal erosion (parental material) and river input (sediment trap) resides within the matrix-free, LD fraction (i.e., not mineral-associated), predominantly in the form of large vascular plant debris. By contrast, shelf sediments from the mid-shelf (at water depths >30 m, ~40 km from the coast) collectively show that most of the terrOC is present in the fine HD fraction (<38 μm) as mineral-associated OC, while LD material accounts on average only about 10% of OC. It is likely that the OC of the shelf LD fraction is not purely of terrestrial origin since δ$^{13}$C–OC values across the land-ocean transect show mild isotopic enrichment in mid-shelf samples (from −26.2‰ in parent material to −25.9‰ on the mid-shelf, see Table 1 and supplementary information Fig. 2), indicating that a part of OC in the shelf LD fraction has a marine origin. This means that the decreasing trend in plant debris abundance with increasing water depth may even be stronger. In short, the consistent disappearance of LD-OC implies that a large fraction of it is lost during transport from its source on land to its sink on the shelf.

SEM images (Fig. 3) demonstrate that, in the shallow coastal zone at 0–2 and 2–5 m water depth, visible plant fragments are abundant in the HD > 63 μm fractions (Fig. 3). As a result, the OC content of coarse HD

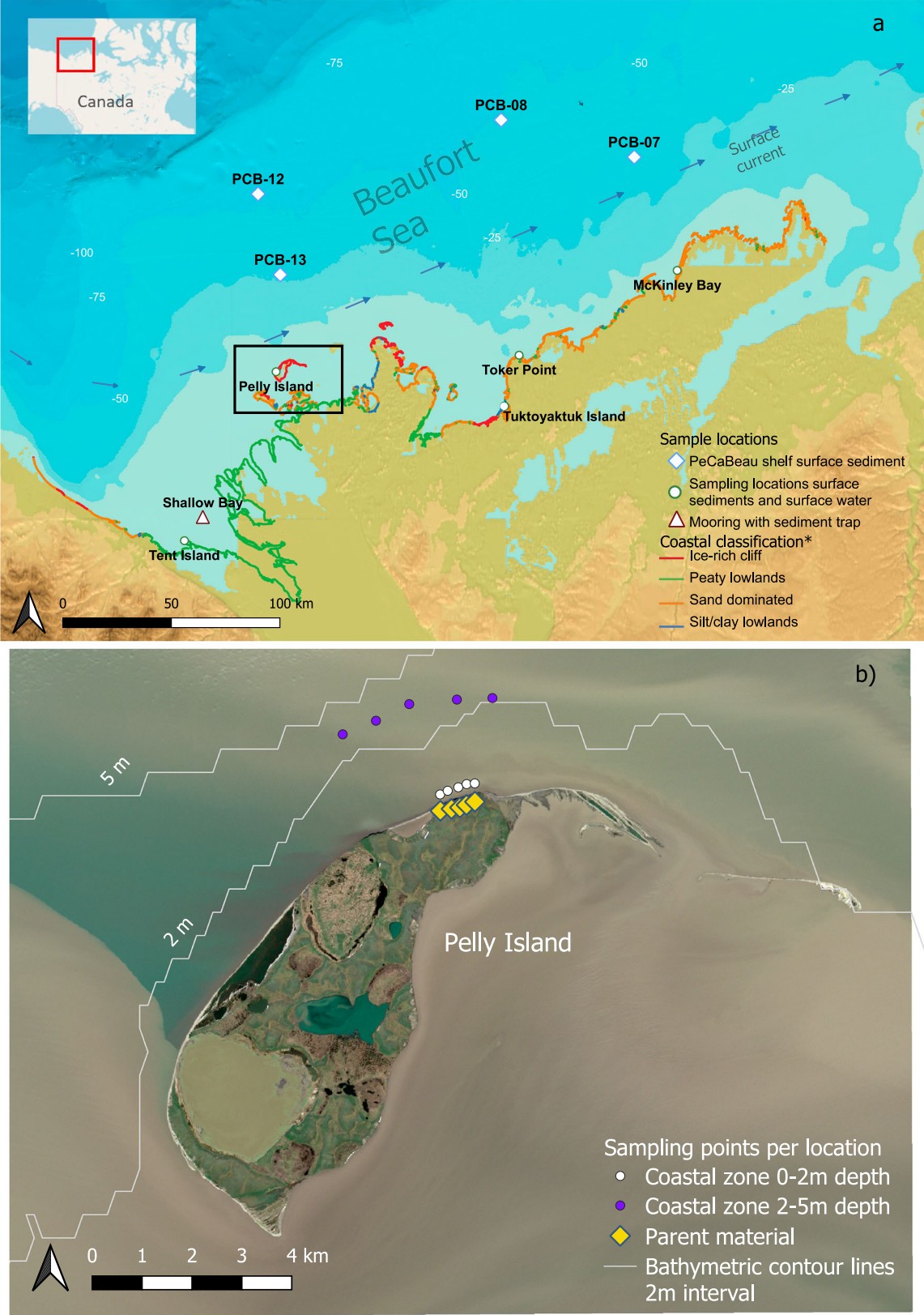

**Fig. 1 | Sampling locations in the Canadian Beaufort Sea.** Samples were collected between 2021 and 2023 during multiple field campaigns. **a** Overview map of all sampled locations, including the sediment trap at Shallow Bay (red triangle) and surface sediments collected during the PeCaBeau cruise 27 (blue diamond), together with additional sampling stations (green circles). **b** Sampling strategy across different zones used in this study, including permafrost sites (yellow diamond), shallow waters at 0–2 m depth (white circles), and waters at 2–5 m depth (purple circles). Bathymetry is shown as white contour lines retrieved from GEBCO, and background satellite imagery was obtained from Sentinel Hub[28,61].

**Table 1 | Sample locations and organic carbon characteristics of different material types**

| Sampling location | Coastal type* | Location | | Weight partitioning (%) | | | | | Organic carbon partitioning (%) | | | | | OC (%) | | | | | δ¹³C (‰) | | | | |
|---|---|---|---|---|---|---|---|---|---|---|---|---|---|---|---|---|---|---|---|---|---|---|---|
| | | | | Low density | High density <38 µm | High density 38–63 µm | High density 63–200 µm | High density >200 µm | Low density | High density <38 µm | High density 38–63 µm | High density 63–200 µm | High density >200 µm | Low density | High density <38 µm | High density 38–63 µm | High density 63–200 µm | High density >200 µm | Low density | High density <38 µm | High density 38–63 µm | High density 63–200 µm | High density >200 µm |
| Permafrost parent material* | I | Tent Island | BSC21-TI-PM-2 | 5.09 | 81.0 | 11.0 | 2.46 | 0.50 | 64 | 21.2 | 6.5 | 6.0 | 2.21 | 25.1 | 0.48 | 1.07 | 5.24 | 9.73 | −27.0 | −27.1 | −27.2 | −26.4 | −26.4 |
| | | | BSC21-TI-PM-4 | 4.38 | 82.2 | 11.4 | 1.77 | 0.34 | 58 | 26.1 | 4.3 | 7.9 | 3.44 | 21.9 | 0.57 | 0.69 | 6.66 | 14.6 | −27.1 | −27.4 | −27.6 | −27.4 | −26.7 |
| | III | Pelly Island | BSC-21-PEL-PM-2 | 1.36 | 73.6 | 9.32 | 13.2 | 2.48 | 55.9 | 37.8 | 2.96 | 2.87 | 0.43 | 35.7 | 0.48 | 0.22 | 0.23 | 0.19 | −25.8 | −26.5 | −26.8 | −26.4 | −26.0 |
| | | | BSC-21-PEL-PM-4 | 1.01 | 81.6 | 8.98 | 6.99 | 1.42 | 42.4 | 49.5 | 3.42 | 4.06 | 0.62 | 38.3 | 0.89 | 0.36 | 0.46 | 0.39 | −25.9 | −26.4 | −26.5 | −26.5 | −26.2 |
| | | | BSC-22-PEL-PM-1 | 1.05 | 80.6 | 8.90 | 7.41 | 2.07 | 40.5 | 53.7 | 3.36 | 2.01 | 0.37 | 35.3 | 0.93 | 0.34 | 0.22 | 0.22 | −26.4 | −26.6 | −26.7 | −26.5 | −27.4 |
| | | | BSC-22-PEL-PM-2 | 3.98 | 21.5 | 40.1 | 31.3 | 3.07 | 41.1 | 51.1 | 4.54 | 2.80 | 0.47 | 30.4 | 0.75 | 0.39 | 0.26 | 0.24 | −26.5 | −26.8 | −26.8 | −26.8 | −27.7 |
| | II | Tuktoyaktuk Island | BSC-21-TuKI-PM-1 | 0.82 | 39.3 | 15.3 | 30.7 | 13.9 | 44.2 | 33.4 | 5.41 | 14.6 | 2.37 | 40.3 | 0.57 | 0.28 | 0.40 | 0.26 | −25.3 | −26.6 | −26.8 | −27.6 | −28.1 |
| | | | BSC-21-TuKI-PM-2 | 0.95 | 55.1 | 15.7 | 27.8 | 0.44 | 40.3 | 36.3 | 4.68 | 15.0 | 3.74 | 34.9 | 0.59 | 0.23 | 0.39 | 0.39 | −25.9 | −26.5 | −27.0 | −27.4 | −27.7 |
| | IV | Toker Point | BSC21-TP-PM-2 | 2.83 | 8.11 | 2.71 | 65.8 | 20.6 | 82 | 7.7 | 0.59 | 8.2 | 1.77 | 30.3 | 1.00 | 0.23 | 0.13 | 0.09 | −26.8 | −25.7 | −26.4 | −26.6 | −27.2 |
| | | | BSC21-TP-PM-4 | 0.25 | 12.8 | 7.86 | 26.6 | 52.4 | 29.4 | 48.8 | 3.89 | 9.88 | 8.10 | 38.4 | 1.23 | 0.16 | 0.12 | 0.05 | −25.5 | −26.2 | −26.4 | −27.3 | −27.5 |
| | | McKinley Bay | BSC21-MK-PM-2 | 17.0 | 4.47 | 2.57 | 46.6 | 29.5 | 93 | 2.6 | 0.63 | 2.9 | 1.1 | 37.5 | 3.97 | 1.69 | 0.43 | 0.25 | −27.1 | −27.3 | −27.4 | −27.1 | −27.2 |
| | | | BSC21-MK-PM-4 | 0.24 | 7.10 | 2.71 | 43.1 | 46.9 | 7.92 | 13.6 | 7.38 | 41.8 | 29.3 | 10.6 | 0.61 | 0.87 | 0.31 | 0.20 | −25.3 | −25.4 | −27.0 | −26.7 | −26.8 |
| Median | | | | 1.21 | 47.2 | 9.15 | 27.2 | 2.77 | 43.3 | 34.9 | 4.11 | 6.93 | 1.99 | 35.1 | 0.68 | 0.35 | 0.35 | 0.25 | −26.2 | −26.6 | −26.8 | −26.7 | −27.2 |
| IQR | | | | 3.43 | 71.6 | 10.3 | 33.1 | 26.5 | 22.4 | 33.9 | 2.17 | 10.5 | 3.16 | 11.7 | 0.41 | 0.60 | 0.23 | 0.20 | 1.35 | 0.78 | 0.57 | 0.91 | 1.18 |
| 0–2 m depth | I | Tent Island | BSC21-TI-RZ-01 | 64.7 | 27.4 | 7.85 | 0.03 | <0.01 | 50.4 | 20.1 | 15.2 | 14.3 | <0.01 | 18.2 | 0.59 | 0.91 | 2.90 | 15.2 | −26.0 | −26.5 | −26.4 | −26.5 | −26.5 |
| | | | BSC21-TI-RZ-03 | 53.7 | 35.8 | 10.5 | <0.01 | 0.02 | 67.5 | 13.8 | 9.26 | 9.46 | <0.01 | 24.8 | 0.41 | 0.56 | 1.95 | – | −26.2 | −26.7 | −26.8 | −27.6 | – |
| | III | Pelly Island | BSC22-PEL-RZ-01-wading | 0.26 | 1.02 | 0.25 | 6.58 | 91.9 | 32.5 | 10.7 | 0.72 | 4.36 | 51.7 | 24.9 | 1.26 | 0.22 | 0.16 | 0.11 | −26.3 | −26.9 | −27.8 | −27.1 | −27.6 |
| | | | BSC22-PEL-RZ-03-wading | 0.25 | 2.25 | 1.01 | 4.22 | 92.3 | 13.9 | 13.4 | 0.76 | 6.32 | 65.7 | 10.4 | 1.55 | 0.23 | 0.22 | 0.14 | −26.8 | −26.7 | −27.5 | −27.1 | −28.4 |
| | III | Tuktoyaktuk Island | BSC21-TUK-RZ-01 | 1.56 | 2.43 | 1.39 | 47.8 | 46.9 | 34.3 | 9.93 | 3.96 | 32.2 | 19.6 | 13.4 | 1.01 | 0.42 | 0.12 | 0.21 | −25.3 | −26.6 | −27.1 | −27.2 | −28.0 |
| | | | BSC21-TUK-RZ-03 | 0.16 | 3.68 | 4.41 | 85.8 | 5.98 | 8.36 | 7.27 | 2.47 | 48.6 | 33.3 | 3.24 | 0.74 | 0.26 | 0.18 | 0.36 | −26.2 | −26.7 | −27.1 | −27.7 | −28.7 |
| | IV | Toker Point | BSC21-TP-RZ-01 | 0.17 | 1.75 | 3.75 | 61.0 | 33.3 | 2.14 | 14.4 | 4.03 | 61.6 | 17.8 | 1.03 | 0.83 | 0.13 | 0.09 | 0.07 | −26.2 | −25.9 | −26.9 | −27.6 | −28.6 |
| | | | BSC21-TP-RZ-02 | 0.16 | 2.42 | 6.31 | 70.1 | 21.0 | 4.03 | 30.7 | 5.23 | – | 60.1 | 1.19 | 1.08 | 0.10 | – | 0.14 | −26.9 | −25.7 | −26.9 | −27.5 | −28.3 |
| | | McKinley Bay | BSC21 MK-RZ-1 | 0.05 | 0.31 | 0.02 | 55.3 | 44.3 | 0.72 | 3.24 | – | 43.3 | 52.8 | 1.16 | 0.89 | – | 0.07 | 0.07 | – | −25.1 | – | −26.0 | −28.3 |
| | | | BSC21 MK-RZ-3 | 0.08 | 0.23 | <0.01 | 48.7 | 51.0 | – | 3.65 | – | 47.8 | 48.6 | – | 0.88 | – | 0.06 | 0.06 | −26.6 | −25.0 | – | −25.6 | −27.7 |
| Median | | | | 0.21 | 2.33 | 3.75 | 48.7 | 45.6 | 11.1 | 12.1 | 4.00 | 32.2 | 50.1 | 10.4 | 0.88 | 0.24 | 0.16 | 0.14 | −26.2 | −26.5 | −27.0 | −27.2 | −28.3 |
| IQR | | | | 14.4 | 8.76 | 6.45 | 60.1 | 57.6 | 36.5 | 9.5 | 7.07 | 40.3 | 35.2 | 20.3 | 0.43 | 0.38 | 1.01 | 0.21 | 0.64 | 1.11 | 0.60 | 1.24 | 0.79 |

**Table 1 (continued) | Sample locations and organic carbon characteristics of different material types**

| Sampling location | Coastal type* | Location | Sample | Weight partitioning (%) | | | | | Organic carbon partitioning (%) | | | | | OC (%) | | | | | δ¹³C (‰) | | | | |
|---|---|---|---|---|---|---|---|---|---|---|---|---|---|---|---|---|---|---|---|---|---|---|---|
| | | | | Low density | High density <38 μm | High density 38–63 μm | High density 63–200 μm | High density >200 μm | Low density | High density <38 μm | High density 38–63 μm | High density 63–200 μm | High density >200 μm | Low density | High density <38 μm | High density 38–63 μm | High density 63–200 μm | High density >200 μm | Low density | High density <38 μm | High density 38–63 μm | High density 63–200 μm | High density >200 μm |
| | I | Tent Island | Surface water | 7.41 | 88.3 | 2.73 | 0.81 | 0.80 | 50.9 | 37.4 | 0.80 | 2.72 | 8.10 | 23.5 | 1.45 | 1.01 | 11.6 | 34.5 | −26.7 | −27.4 | −27.6 | −27.0 | −26.9 |
| | III | Pelly Island | | 1.28 | 90.5 | 4.22 | 3.97 | − | 22.9 | 76 | 0.50 | 0.44 | − | 29.3 | 1.37 | 0.19 | 0.18 | − | −26.5 | −26.9 | −27.1 | −27.2 | − |
| | II | Tuktoyaktuk Island | | 0.41 | 99.6 | − | − | − | 13.5 | 87 | − | − | − | 42.9 | 1.14 | − | − | − | −26.1 | −27.4 | − | − | − |
| | IV | Toker Point | | 3.13 | 73.0 | 16.0 | 5.74 | 2.13 | 21.75 | 77 | 0.55 | 0.41 | − | 13.2 | 2.02 | 0.07 | 0.14 | − | −27.4 | −27.9 | −27.8 | −28.4 | − |
| | | McKinley Bay | | 0.02 | 3.05 | 0.22 | 70.7 | 26.0 | 1.65 | 38 | 0.09 | 60.1 | − | 13.2 | 2.02 | 0.07 | 0.14 | − | −27.4 | −27.9 | −27.8 | −28.4 | − |
| | Median | | | 1.28 | 88.3 | 3.48 | 4.86 | 2.13 | 21.7 | 76.1 | 0.53 | 1.58 | 8.10 | 23.5 | 1.45 | 0.13 | 0.16 | 34.5 | −26.7 | −27.4 | −27.7 | −27.8 | −26.9 |
| | IQR | | | 5.05 | 57.0 | 12.2 | 52.9 | 25.2 | 29.4 | 44.1 | 0.55 | 45.4 | − | 22.9 | 0.76 | 0.74 | 8.57 | − | 1.06 | 0.77 | 0.61 | 1.42 | − |
| 2.2 m depth | | Shallow Bay | Sediment trap | 4.01 | 71.6 | 18.7 | 5.71 | − | 55.7 | 32.5 | 6.57 | 5.21 | − | 23.6 | 0.77 | 0.60 | 1.55 | − | −26.7 | −27.1 | −27.8 | −27.4 | − |
| 2–5 m depth | I | Tent Island | BSC21-TI-RZ-1 | 34.2 | 0.11 | 14.5 | 1.82 | 49.4 | 0.38 | 7.72 | 61.0 | 24.1 | 6.84 | 29.5 | 0.67 | − | 0.26 | 9.71 | −25.9 | −27.3 | −27.6 | −28.0 | −26.6 |
| | I | | BSC21-TI-RZ-5 | 28.5 | − | 31.4 | 4.01 | 36.1 | − | 11.6 | 54.8 | 25.1 | 8.58 | 22.2 | 0.58 | − | 0.27 | − | −26.8 | −27.3 | −27.8 | −27.9 | − |
| | III | Pelly Island | BSC22-PEL-RZ-1-by boat | 1.24 | 1.68 | 94.9 | 0.25 | 1.92 | 0.31 | 9.08 | 8.6 | 79.8 | 2.19 | 5.54 | 0.70 | − | 0.18 | 0.25 | −26.0 | −26.9 | −27.1 | −26.8 | −27.0 |
| | III | | BSC22-PEL-RZ-3-by boat | 8.90 | 0.11 | 33.5 | 0.25 | 57.3 | 0.17 | 11.8 | 30.4 | 53.8 | 3.84 | 20.9 | 0.51 | − | 0.33 | 0.14 | −26.2 | −27.4 | −27.6 | −27.3 | −26.9 |
| | III | | BSC21 PEL-DZ-1 wading | 2.15 | − | 0.12 | 0.26 | 97.5 | − | − | 28.2 | 63.9 | 7.95 | 16.7 | 0.52 | − | 0.59 | − | −26.1 | −27.4 | −27.7 | − | − |
| | III | | BSC21 PEL-DZ-3 wading | 8.75 | − | 6.22 | 1.76 | 83.3 | − | 29.6 | 23.9 | 39.7 | 6.81 | 22.7 | 0.52 | − | 0.81 | − | −26.7 | −27.3 | −27.1 | −27.1 | − |
| | II | Tuktoyaktuk Island | BSC22-TUK-DZ-1 | 6.12 | 0.15 | 2.71 | 1.87 | 89.1 | − | 5.83 | 27.4 | 59.9 | 6.84 | 29.2 | 0.98 | − | 4.12 | − | −26.5 | −27.4 | −27.2 | −27.0 | −27.9 |
| | II | | BSC22-TUK-DZ-3 | 0.91 | 0.06 | 0.19 | 1.66 | 97.2 | − | 2.54 | 34.9 | 60.8 | 1.78 | 31.9 | 0.85 | − | 0.92 | − | −26.5 | −27.3 | −27.0 | −26.9 | − |
| | II | | BSC22-TUK-DZ-1-SB | 3.15 | <0.01 | 0.53 | 1.34 | 95.0 | 0.25 | 0.94 | 30.6 | 67.3 | 0.90 | 25.5 | 0.86 | − | 0.42 | 5.79 | −26.4 | −27.2 | −26.8 | −26.2 | − |
| | II | | BSC22-TUK-DZ-2-SB | 0.49 | − | 0.07 | 0.98 | 98.5 | − | 6.78 | 33.5 | 47.1 | 12.6 | 30.1 | 0.65 | − | 6.42 | − | −26.3 | −27.2 | −26.6 | −26.9 | − |
| | IV | Toker Point | BSC21 TP-DZ-1 | 0.02 | 67.3 | 32.4 | 0.03 | 0.19 | − | 75.5 | − | 10.6 | 13.9 | − | 1.07 | − | − | 0.07 | − | −26.6 | − | −27.4 | −27.9 |
| | IV | | BSC21 TP-DZ-3 | 0.11 | 78.5 | 21.2 | 0.04 | 0.16 | − | 66.8 | − | 15.4 | 17.8 | − | 1.52 | − | − | 0.09 | − | −26.8 | − | −26.9 | −28.2 |
| | | McKinley Bay | BSC21 MK-DZ-1 | 0.03 | 28.8 | 70.6 | 0.17 | 0.41 | − | 27.9 | 1.45 | 9.80 | 60.9 | 0.52 | 1.27 | − | − | 0.06 | −26.1 | −24.0 | − | −26.2 | −28.6 |
| | | | BSC21 MK-DZ-3 | 0.03 | 22.3 | 76.9 | 0.24 | 0.55 | − | 23.4 | 1.65 | 10.7 | 64.2 | 0.59 | 1.38 | − | − | 0.07 | −25.1 | −23.7 | − | −26.7 | −28.2 |
| | Median | | | 1.70 | 1.68 | 17.8 | 0.62 | 53.3 | 0.25 | 11.6 | 29.3 | 43.4 | 6.83 | 11.9 | 22.5 | 0.78 | 0.51 | 0.13 | −26.3 | −27.3 | −27.1 | −26.9 | −27.9 |
| | IQR | | | 8.70 | 48.0 | 42.3 | 1.54 | 95.0 | 42.3 | 21.1 | 22.1 | 47.4 | 6.01 | 56.8 | 21.1 | 0.56 | 1.45 | 4.33 | 0.49 | 0.58 | 0.69 | 0.59 | 1.31 |
| | I | Tent Island | Surface water | 9.83 | 0.04 | 0.34 | 2.58 | 87.2 | 0.34 | − | 23.0 | 52.9 | 21.5 | 0.47 | 26.5 | 1.80 | 6.51 | 33.1 | −27.0 | −27.4 | −26.9 | −26.9 | −26.5 |
| | III | Pelly Island | | 0.34 | − | 25.8 | 1.99 | 98.0 | − | − | 25.8 | 74.2 | − | − | 19.7 | 1.15 | − | − | −26.2 | −27.2 | − | − | − |
| | II | Tuktoyaktuk Island | | 0.07 | − | 0.23 | 0.10 | 99.4 | 0.07 | − | 0.23 | 99.4 | 0.31 | − | 9.07 | 4.07 | 3.16 | − | −26.8 | −27.5 | − | −27.6 | − |

**Table 1 (continued) | Sample locations and organic carbon characteristics of different material types**

| | Sample | | WP | OC | OCp | | | | | | | | | | | | | | δ¹³C | | | |
|---|---|---|---|---|---|---|---|---|---|---|---|---|---|---|---|---|---|---|---|---|---|---|
| IV | Toker Point | | – | 100 | – | – | – | – | – | – | 2.58 | – | – | 19.7 | – | 33.1 | – | – | 2.58 | – | –28.4 | – | –26.5 |
| | McKinley Bay | | – | 100 | – | – | – | – | – | – | 2.58 | – | – | 17.4 | – | – | – | 1.85 | – | – | –28.4 | – | –26.5 |
| Median | | | 1.99 | 99.4 | 5.11 | 0.21 | 0.04 | 23.0 | 10.9 | 1.10 | 2.58 | 4.84 | 19.7 | 9.87 | 33.1 | – | 2.58 | – | –27.5 | –27.2 | –26.8 | –27.3 | –26.5 |
| IQR | | | 2.47 | 7.39 | – | – | – | 25.5 | 36.5 | – | 1.85 | – | 17.4 | – | – | – | 1.10 | – | 1.10 | 0.74 | – | – | – |
| No classification | PCB13 | 30 m depth | 1.09 | 98.6 | 0.12 | 0.05 | 0.15 | 17.0 | 9.71 | 0.37 | 1.14 | 3.03 | 21.4 | 3.53 | 3.19 | 0.11 | 1.14 | –27.1 | –24.9 | –26.8 | –26.3 | –26.5 |
| | PCB7 | 40 m depth | 0.56 | 95.5 | 1.63 | 0.38 | 1.95 | 9.71 | 27.0 | 0.51 | 1.44 | 0.70 | 27.0 | 1.16 | 2.11 | 0.51 | 1.03 | –25.8 | –25.9 | –25.8 | –25.2 | –20.7 |
| | PCB8 | 50 m depth | 0.64 | 89.5 | 2.36 | 1.46 | 6.00 | 10.3 | 22.0 | 0.92 | 1.77 | 0.77 | 22.0 | 0.86 | 0.86 | 0.92 | – | –25.8 | –25.7 | –25.8 | –25.8 | –22.0 |
| | PCB12 | 55 m depth | 0.72 | 98.8 | 0.18 | 0.33 | – | 12.3 | 23.9 | 0.21 | 1.23 | 2.14 | 23.9 | 0.90 | – | – | – | –26.5 | –26.1 | –26.5 | –26.2 | –22.0 |
| Median | | | 0.68 | 97.0 | 0.91 | 1.14 | 1.14 | 11.3 | 22.9 | 0.50 | 1.27 | 1.45 | 22.9 | 1.03 | 2.11 | 0.51 | 2.11 | –26.2 | –25.9 | –25.9 | –26.0 | –22.0 |
| IQR | | | 0.42 | 7.70 | 2.05 | 4.79 | 1.42 | 6.00 | 4.39 | 1.44 | 0.24 | 2.10 | 4.70 | 2.41 | 2.32 | 0.81 | 1.44 | 0.94 | 2.32 | 1.14 | 0.61 | 0.90 | 5.77 |

Samples include permafrost parent material, surface sediments, shelf sediments, surface water, and sediment trap material. For each sample type, the median value and interquartile range (IQR) are reported. Median values are shown instead of means, as coastal zones can cause large variability between samples. Reported parameters include weight partitioning (%), OC content (%), organic carbon (OC) partitioning (%), and δ¹³C (‰). Not all measurements were possible for every location due to limited sample availability. Yellow shaded values in the OC (%) column indicate high-density (HD) samples with OC content greater than 1%, also visualized in Supplementary Information 4. Coastal types, indicated with an asterisk (*), are categorized as: (I) low-lying peat, (II) fine clay-rich with organic active layer, (III) fine clay-rich with overlying OC-rich material, and (IV) sandy dunes with organic active layer. Further details on coastal type classification are provided in van Crimpen et al.[22]

material, which is typically low (below 1%), exhibits occasionally anomalously high OC values because of the presence of carbon-rich plant debris (Supplementary information Fig. 4). This is further corroborated by occasionally relatively high C/N ratios, which are normally high for plant material but not for reworked, soil-derived OC[42] (Supplementary information Fig. 3). The reason that the HD coarse material (>63 μm) contains plant debris could be due to incomplete (density/size) fractionation. Alternatively, these large plant fragments trapped in the shallow zone (0-5 m) could have a high density due to factors such as heavy waterlogging of plant debris, or incorporation of small inorganic particles within the cell walls, including silica (opal) structures produced in and between the cells of plants that can change the original tissue density[43]. However, to the best of our knowledge, the impact of particle or silica inclusion on the density of plant fragments has not been addressed in the literature.

Regardless of whether plant fragments are in the LD or coarse HD fractions, they dominate the 0–5 m zone and are virtually absent on the shelf. How this plant debris is trapped in the coastal region is counter-intuitive as, being relatively light, this OC-rich fraction should be further transported offshore by currents and wave-related resuspension[20] However, because of its large size (Fig. 3)—i.e., within the range of coarse sandy material—its settling velocity is high despite the relatively low density (Fig. 5). This is because particle diameter (i.e., size) has a larger effect on particle transport than density. In fact, according to Stokes' law, settling velocity is proportional to the square of particle size but only the first power of excess density[17] This means that a large range in diameter (several orders of magnitude) will be more important than a small range in density[17,18] Following this mechanism, the finer—but denser—particles being rich in mineral-associated OC are transported further out of the 0-5 m depth zone towards the mid-shelf[18,21,26,44]. Furthermore, SEM images indicate that the plant debris in the LD fraction decreases in size when moving further to the shelf (Fig. 3), which might indicate either mechanical breakdown, degradation, or selective transport of the finer (more mobile) plant debris. Additionally, incorporation of sediments into ice during freeze-up in early fall may transport a portion of it offshore[45]. This may be particularly relevant for sediment released via coastal erosion, as this process is at its peak during late summer and early fall, at the same time when ice formation starts. On the Chukchi and Beaufort shelves, it is currently estimated that 5–8 Tg of eroded sediments become incorporated into the sea ice annually[46], representing approximately 6.5–10% of the total 78 Tg sediment released annually by coastal erosion in that same area[38,47] These sediments, and their OC, can then be transported offshore and released during melting, potentially far from the source[48–50]. While a few studies suggest most of this sediment is fine-grained[48,51,52], we cannot rule out the inclusion of OC-rich vascular plant fragments that are dominant in our study. We therefore suggest that the stark decrease of the OC-rich LD fraction is likely a combination of mechanical and biological breakdown, combined with transport processes offshore.

To explore cross-shelf transport trends at a pan-Arctic scale, we combined our results with data obtained for the Siberian margin by Tesi et al.[18] and for the Canadian Beaufort coastal region by van Crimpen et al.[22], primarily focusing on the LD fraction. Previously, van Crimpen et al.[22] showed that along the Canadian Beaufort Sea coast up to 52% of OC in permafrost parent material resides in the matrix-free LD fraction, which implies that this shallow coastal zone may account for >50% of terrestrial OC cycling in the Arctic Ocean. Similarly, Schreiner et al.[27] found that 40 to 60% of the OC within 2.5 m water depth in the Simpson Lagoon (Alaska), off the Colville River, is associated with particles lighter than 2 g/cm³, consistent with our observations. Furthermore, Tesi et al.[18] observed a similar trend along the Siberian margin, where plant debris concentration decreased with increasing water depth. However, the observed spatial trends were less pronounced, likely due to the study's focus on deeper waters that excluded the shallowest (<5 m) regions, where—according to our findings—the most significant changes occur. Ultimately, this suggests that the observed spatial patterns in this study are widespread across Arctic margins. Beyond the Arctic, the importance of shallow coastal zones has been

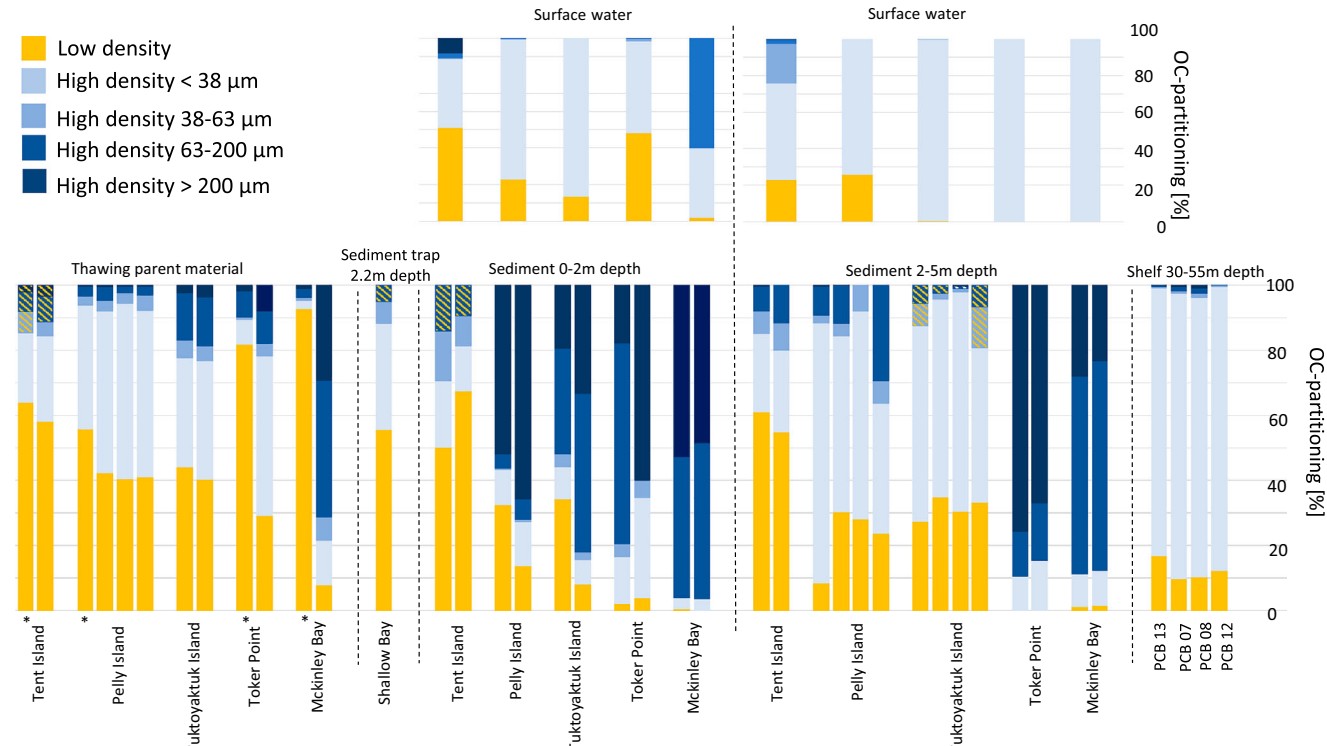

**Fig. 2 | Organic carbon distribution across density fractions of sampled sediments.** Fractionated samples are shown with their corresponding organic carbon (OC) content per fraction (in %). Sampling locations are ordered from land (left) toward the shelf (right), and within each zone from west to east. Active layer samples are indicated with an asterisk (*). Hatched high-density bars mark fractions with OC content >1%. Colors represent sediment density fractions: low density (yellow), high density <38 μm (light blue), high density 38–63 μm (medium blue), high density 63–200 μm (dark blue), and high density >200 μm (very dark blue). Raw data are provided in Table 1 and Supplementary Information 4.

explored in several studies, such as in the central Gulf of Papua, where significant carbon remineralization in sediments was identified, contributing to overall organic carbon loss[53]. Cai[54] expanded on this by discussing the "estuarine and coastal ocean carbon paradox," showing that coastal and estuarine zones may function as sites of carbon incineration, transforming terrestrial inputs into $CO_2$. Even the Amazon–Guianas tropical mobile mud belt is a site of rapid carbon remineralization through microbial degradation[40]

To further contextualize this new dataset obtained for different shelves, we looked at the spatial distribution of surface sediments included in the CASCADE database (Circum-Arctic Sediment CArbon DatabasE[36]. CASCADE encompasses all available data points for organic geochemistry from circum-Arctic Ocean surface sediments collected over several decades. The comparison with CASCADE highlights that the shallow coastal zone is significantly understudied (Fig. 5), accounting for only 6% out of the 2955 surface sediment samples collected on the Arctic shelves (defined as 0–120 m water depth). Most of the samples and data were collected from regions where the LD fraction may account for ca. 10% of the total OC, a figure that may be understated given that part of this LD material is believed to originate from marine sources as previously discussed.

The limited sample intensity in the 0–5 m zone has valid logistical reasons, as expeditions with larger vessels are restricted to deeper waters. However, we show that by disregarding the shallow coastal regions, we distort our view of source-to-sink OC dynamics (Fig. 4). Arctic shelf sediments do not necessarily reflect the type and quantity of terrOC that can be found in erosive coastal parent material. Our findings thus emphasize the need to intensify sampling efforts in these shallow zones in order to gain a better understanding of coastal carbon dynamics[4]. Furthermore, these findings have significant implications for the interpretation of geochemical proxies commonly used to trace land-ocean transport, particularly terrestrial biomarkers[18,27,32,55] Given that biomarker concentrations vary depending on the particle type and size, the observed shallow-water sorting will likely exert a strong influence on the spatial distribution of biomarkers[18,27] For example, lignin phenols may exhibit different spatial patterns compared to terrestrial wax lipids, reflecting their specific affinities for the low-density plant debris fraction versus the mineral-associated fraction.

To fully understand the functioning of pan-Arctic shallow coastal zones, it is necessary to disentangle how degradation and mechanical breakdown interact in specific environmental contexts. Along with hydrodynamic sorting, these key processes determine the fate of terrestrial OC in general, and vascular plant material in particular, which is required for reliable assessments of carbon cycling across the land-ocean interface.

## Methods
### Study area and sample collection
The study area, known for its rapidly eroding coastline, is located on the Indigenous lands of the Inuvialuit Settlement Region along the Beaufort Sea coast, including the Mackenzie Delta estuary in the Northwest Territories and Yukon, Canada. It spans from King Point in the west to McKinley Bay in the east[22] (Fig. 1). This coastline, which is part of the continuous permafrost zone, is 5672 km long and accounts for 5.6% of the Arctic coastline[56,57]

In van Crimpen et al.[22] 2500 km of coastline along the Canadian Beaufort Sea coast was classified into four geomorphological types based on data from the Coastal Information System (CIS)[58] and observations in the field. Type I coasts consist of low-lying grassy peatlands with high organic carbon (OC) and low erosion rates. These coasts are primarily found in the western Mackenzie Delta, such as Tent Island, and cover approximately 46% of the study area. Type II coasts are characterized by clay-rich material, medium to high ground-ice content, and moderate backshore elevations of about 15 m. Examples include Tuktoyaktuk Island. Type II accounts for

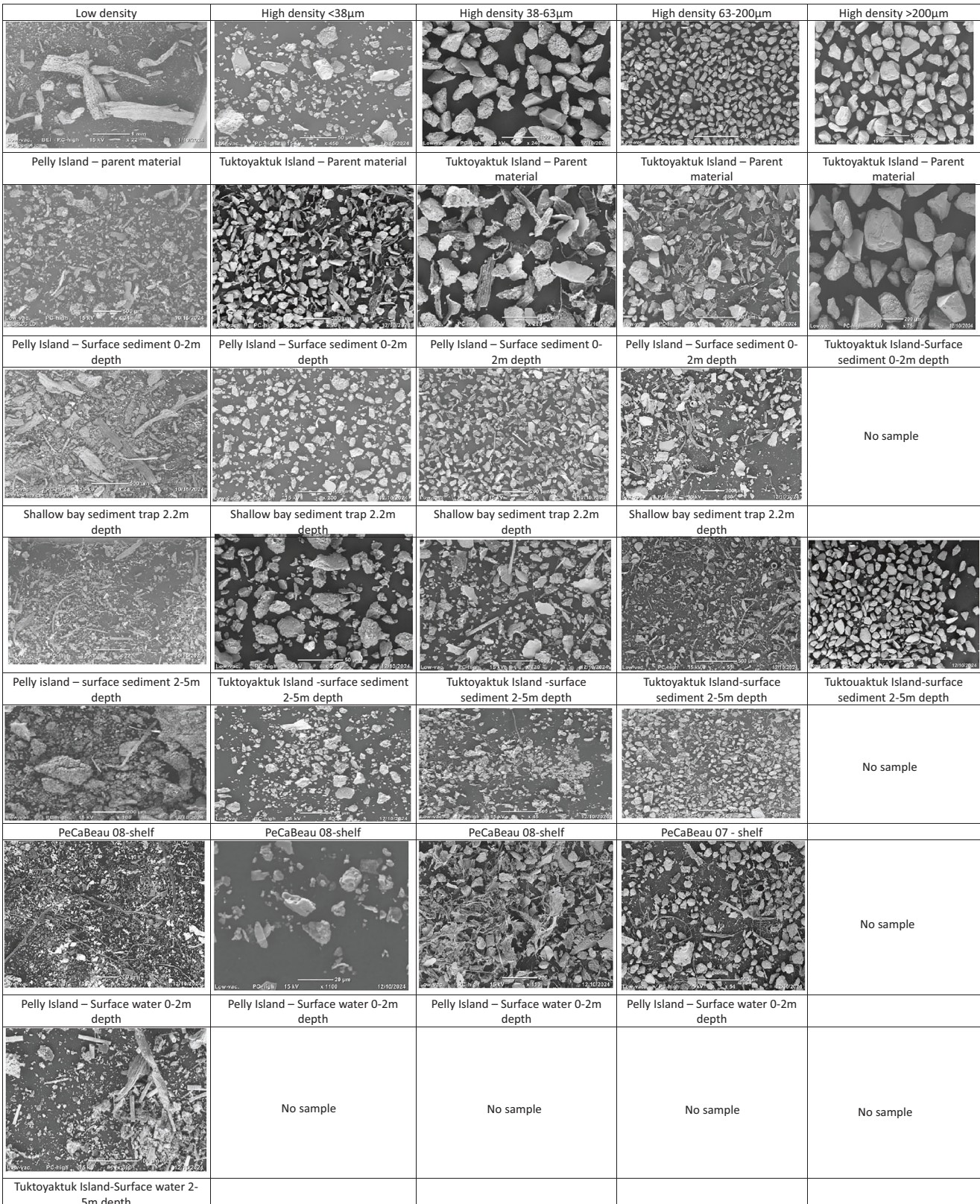

**Fig. 3 | SEM images illustrating the redistribution of the low-density material into the coarser fractions upon entering the marine system.** Parental material (column 1) and Surface sediment samples for both the 0–2 m zone (column 2), the 2–5 m zone (column 3), and the shelf (column 4). For each zone, an image of each fraction is shown, within row 1, the Low density, High density <38 μm (row 2), High density 38–63 μm (row 3), High density 63–200 μm (row 4), and the High density >200 μm (row 5). Scales vary between zones or fractions due to smaller fragment sizes towards the shelf. Additional SEM images can be found in the supplementary information.

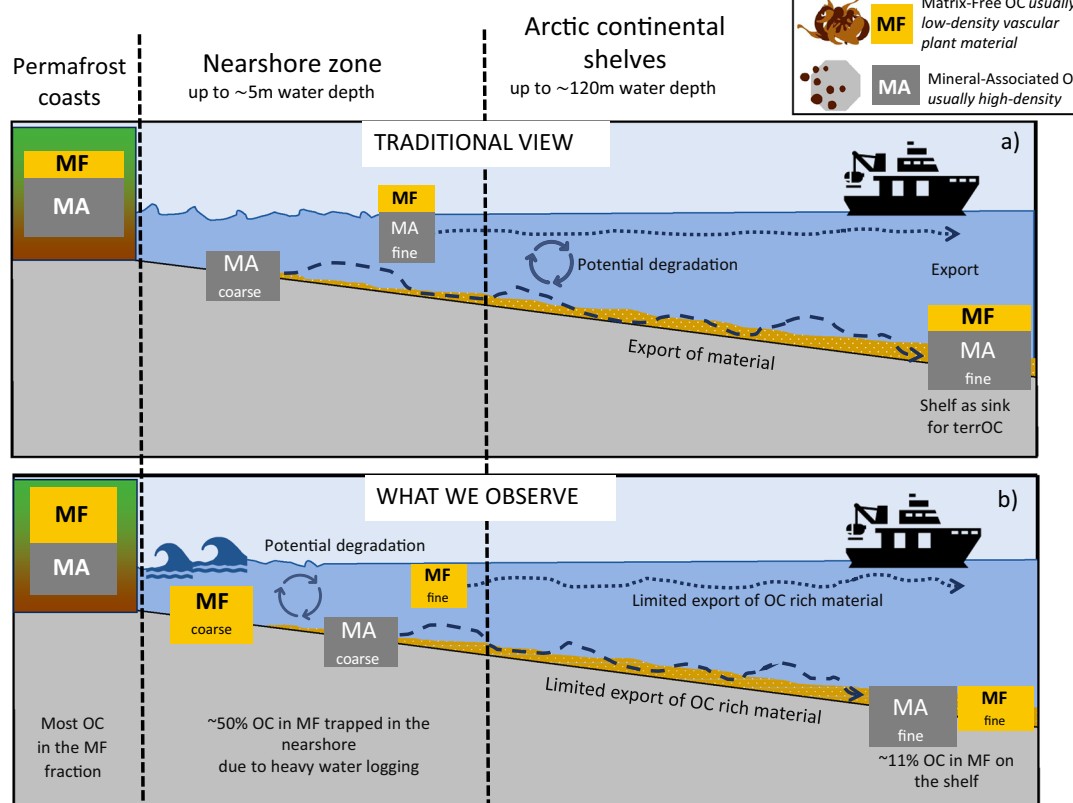

**Fig. 4 | Conceptual diagrams of organic carbon transport in coastal areas.**
**a** traditional view of OC (organic carbon) transport in coastal areas, as well as **b** what we observe. The traditional view represents a source-to-sink transport mechanism, where terrestrial organic carbon (terrOC) is transported from land to marine environments mostly in the mineral-associated (MA) form. Most coarse material remains in the nearshore zone, while finer material is partly degraded and exported to the shelf. In our study, we find that most of the OC resides in the matrix-free (MF) form (OC-partitioning), as large vascular plant material, which becomes water-saturated, increasing its density and causing it to behave as high-density, coarse material. This OC-rich, high-density, large-size material becomes trapped in the nearshore zone, being exposed to resuspension and degradation. Only a limited portion of MF material is exported to the shelf. The presence and behavior of OC-rich MF material challenge the conventional understanding of OC transport and highlight the importance of the dynamic nearshore zone in OC distribution and deposition.

around 4% of the coastline. Type III coasts are high cliffs with an average backshore elevation of approximately 20 m, high ground-ice content, and large retrogressive thaw slumps such as Pelly Island, covering about 8% of our study area. Lastly, Type IV coasts consist of sandy dunes overlain by thick (~3 m) OC-rich layers with low ground-ice content. These coasts are more abundant in the eastern region, such as McKinley Bay and Toker Point, and cover around 42% of the coastline. For more detailed descriptions of these coastal types, see van Crimpen et al.[22]

Based on this coastal classification, eight coastal sampling locations were selected to represent larger coastal areas based on cliff height, ground-ice content, erosion rates, and geomorphological features[22]. To fully trace the land-to-ocean continuum in this study, we additionally sampled coastal surface sediments and surface water at five locations. Here, surface sediment samples were taken in two depth zones (0–2 and 2–5 m) to capture spatial variability and transport mechanisms. Up to a depth of ~1 m, this was done by wading and between 2–5 m with an inflatable boat. For most sites, five samples (300–500 g each) were collected per zone at horizontal intervals of ~25 m. Sampling was conducted using a stainless steel Van Veen grab sampler. Subsamples were extracted with a stainless-steel spoon, stored in Whirl-Pak bags, and frozen at −18 °C until laboratory processing. Suspended particulate material (SPM) was collected from the 0–2 m (~50 L) and the 2–5 m depth (~80 L) zones using pre-rinsed 5 L wine bags. Surface water was filtered through polyether sulfone (PES, 0.45 μm pore size) membranes using a stainless-steel filtration setup. Filters were stored frozen, then thawed, rinsed with Milli-Q water, freeze-dried, weighed, and stored for subsequent analysis.

In addition, a sample from a previously deployed sediment trap was used. The sediment trap was deployed in Shallow Bay (68.9700°N, 136.4424°W) at a water depth of ~2 m from June 30 to September 23, 2022. The trap consisted of a 9 cm diameter PVC pipe fixed vertically to an aluminum tripod equipped with oceanographic sensors. The system was moored with a weighted line connected to a buoy. Details on the setup and deployment can be found in ref. 59.

The four shelf sediment samples (PCB 7, PCB 8, PCB 12, and PCB 13) were collected for the Permafrost Carbon in the Beaufort Sea (PeCaBeau) project, conducted during September–October of 2021 aboard the Canadian Coast Guard vessel *CCGS Amundsen*[28]. Short sediment cores were collected using a Multicoring system and subsampled at 1 cm resolution. For this study, only the first two intervals were used. Subsamples were stored frozen until further analysis.

### Hydrodynamic fractionation of surface and suspended sediment
Surface sediment (ca. 15 g) and suspended particulate material were fractionated based on density and grain size by using an aqueous solution of sodium polytungstate. Freeze-dried samples were subsequently sieved over a mesh of 38, 63, and 200 μm pore size following the methods of refs. 17,18,21,22. Recovery rates of sieved material range from 81.9% to 96.5%, and are consistent with previous studies[22,27,29]

### Geochemical analyses
Surface sediments and suspended particulate matter were weighed, freeze-dried, fractionated (see above), and homogenized before further chemical

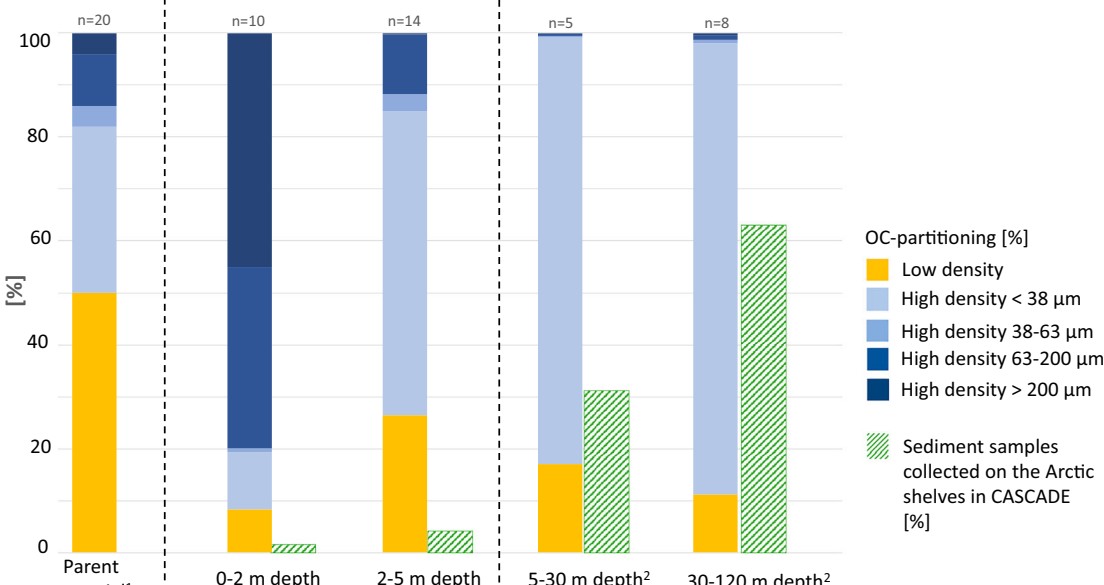

**Fig. 5 | Organic carbon distribution along the Arctic land-ocean continuum reveals the critical role of shallow coastal zones.** Hatched green bars indicate the pan-Arctic sampling distribution across different depth zones (between 0 and 120 m) based on the CASCADE database[62], showing a systematic under-representation of shallow area sampling. Also shown is the stacked organic carbon (OC)-partitioning (%) among sediment fractions for different depth intervals. The declining OC contributions of the matrix-free LD fraction with increasing depth emphasize the disproportionate contribution of nearshore areas (<5 m depth) to pan-Arctic carbon cycling. 1) Data is available through van Crimpen et al.[22]; additional data is available via Tesi et al.[18].

analyses. Organic carbon (OC, wt%), total nitrogen (TN, wt%), C/N ratio and carbon isotopes $\delta^{13}C$ (‰VPDB), were measured. For OC and $\delta^{13}C$ measurements, subsamples (0.5–15 mg) were finely ground, transferred to a silver capsule and acid-fumigated in a desiccator with concentrated HCl (30%) for 24 h to remove carbonates[60]. The samples were then neutralized and dried with NaOH pellets at 60 °C for 48 h and wrapped in additional tin capsules to aid combustion. Analyses were conducted using a Thermo Electron mass spectrometer at the Institute of Polar Sciences, Bologna, Italy. Results for OC and TN were reported as weight percentages, while stable isotope data were expressed in ‰ relative to the Vienna PeeDee Belemnite (VPDB) standard.

**Scanning electron microscopy (SEM)**
Samples were analyzed using a SEM at the Department of Earth Science at the Vrije Universiteit Amsterdam to examine the characteristics of sediment particles. Prior to analysis, samples were dried and mounted on aluminum stubs using double-sided carbon tape. Particles were imaged with a JEOL Neoscope II JCM-6000 SEM under low vacuum at 15 kV, using back-scattered electron imaging and energy-dispersive X-ray spectroscopy (EDS).

## Data availability
All data generated or analyzed during this study are included in this published article and its Supplementary Information. The dataset generated and analyzed in this study has been submitted to the PANGAEA Data Publisher for Earth and Environmental Science (submission ID: PDI-42309) and will be made publicly available with a DOI prior to final publication. Data used from the CASCADE database can be accessed via https://doi.org/10.5194/essd-13-2561-2021.

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

## Acknowledgements
We would like to thank Angus Robertson, Bay Barry, Obie Anikina, James Keevik, Kendyce Cockney, the PeCaBeau team, Lisa Loseto, and the Aurora Research Institute in Inuvik, Northwest Territories, for their assistance with the field work and organization. This program would not have been possible without the support of the Hamlet of Tuktoyaktuk and the Tuktoyaktuk Community Corporation (TCC). Thanks to Dirk Jong and Kirsi Keskitalo for their support with protocols and methods. Thanks to Roel van Elsas for help with SEM imaging. Thanks to Igor Bely for the opportunity to get samples on board your sailing vessel. We are grateful for the logistical support of Natural Resources Canada's Polar Continental Shelf Program (PCSP). This project is funded through the Dutch Research Council (VI. Vidi.193.100) and falls under NWT Research license: 16897 and ILA land use permit: ILA18TN005. We also acknowledge support and opportunities via the EU Horizon 2020 project Nunataryuk (grant no. 773421). We would like to thank the anonymous reviewers and editors for their contributions to this paper.

## Author contributions
J.E.V. acquired funding and provided resources. J.E.V., F.C.J.v.C. and L.M. co-designed the study. F.C.J.v.C. and L.M. conducted the fieldwork facilitated by D.W. Laboratory analyses were performed by F.C.J.v.C., L.M., J.M.v.G., T.T., and L.B., while K.S. and D.W. carried out the mooring work. M.F. and L.B. collected the PeCaBeau samples. Data analysis was done by F.C.J.v.C., L.M., J.M.v.G., T.T., and L.B., with F.C.J.v.C. preparing the figures. The project was supervised by J.E.V., T.T., and D.W., who also provided critical guidance and input for the manuscript. F.C.J.v.C. wrote the first draft of the manuscript, and J.E.V., T.T., and all co-authors contributed to its revision and approved the final version.

## Competing interests
The authors declare no competing interests.
