## [Transparent Peer Review file · Communications Earth & Environment]

Shallow coastal zones are key mediators in Arctic land-ocean carbon fluxes

Corresponding Author: Ms Fleur van Crimpen

Version 0:

Decision Letter:

Dear Ms van Crimpen,

Please accept our apologies for the delay in sending a decision on your manuscript. Your manuscript titled "Bridging the gap: shallow coastal zones are key mediators in Arctic land-ocean carbon fluxes" has now been seen by 2 reviewers, and we include their comments at the end of this message. They find your work of interest, but some important points are raised. We are interested in the possibility of publishing your study in Communications Earth & Environment, but would like to consider your responses to these concerns and assess a revised manuscript before we make a final decision on publication.

We therefore invite you to revise and resubmit your manuscript, along with a point-by-point response that takes into account the points raised. Please highlight all changes in the manuscript text file.

Please submit your point-by-point responses as a separate file, distinct from your cover letter where you can add responses to the Editors' comments that you do not want to be made available to the reviewers. Word files are preferred. We recommend that any figures, tables or graphs that are included in the response to reviewers are also included in the main article or Supplementary Information.

Please use the following link to submit your revised manuscript, point-by-point response to the referees' comments (which should be in a separate document to any cover letter), a tracked-changes version of the manuscript (as a PDF file) and the completed checklist:

Link Redacted

We hope to receive your revised paper within six weeks; please let us know if you aren't able to submit it within this time so that we can discuss how best to proceed. If we don't hear from you, and the revision process takes significantly longer, we may close your file. In this event, we will still be happy to reconsider your paper at a later date, as long as nothing similar has been accepted for publication at Communications Earth & Environment or published elsewhere in the meantime.

Please do not hesitate to contact us if you have any questions or would like to discuss these revisions further. We look forward to seeing the revised manuscript and thank you for the opportunity to review your work.

Best regards,

Dr Nezha MEJJAD
Editorial Board Member
Communications Earth & Environment

orcid.org/0000-0002-6750-6781

Alice Drinkwater, PhD
Associate Editor
Communications Earth & Environment
Consulting Editor
Communications Sustainability

EDITORIAL POLICIES AND FORMATTING

Editorial Policy: [Policy requirements](https://www.nature.com/documents/nr-editorial-policy-checklist.pdf) (Download the link to your computer as a PDF.)

- Behavioural and social science
- Ecological, evolutionary & environmental sciences
- Life sciences

<https://www.nature.com/documents/nr-reporting-summary.zip>

Furthermore, please align your manuscript with our format requirements, which are summarized on the following checklist: [Communications Earth & Environment formatting checklist](https://www.nature.com/documents/commsj-phys-style-formatting-checklist-article.pdf)

and also in our style and formatting guide [Communications Earth & Environment formatting guide](https://www.nature.com/documents/commsj-phys-style-formatting-guide-accept.pdf).

*** DATA: Communications Earth & Environment endorses the principles of the Enabling FAIR data project (<http://www.copdess.org/enabling-fair-data-project/>). We ask authors to make the data that support their conclusions available in permanent, publically accessible data repositories. (Please contact the editor if you are unable to make your data available).

All Communications Earth & Environment manuscripts must include a section titled "Data Availability" at the end of the Methods section or main text (if no Methods). More information on this policy, is available at <http://www.nature.com/authors/policies/data/data-availability-statements-data-citations.pdf>.

If a community resource is unavailable, data can be submitted to generalist repositories such as [figshare](https://figshare.com/) or [Dryad Digital Repository](http://datadryad.org/). Please provide a unique identifier for the data (for example a DOI or a permanent URL) in the data availability statement, if possible. If the repository does not provide identifiers, we encourage authors to supply the search terms that will return the data. For data that have been obtained from publically available sources, please provide a URL and the specific data product name in the data availability statement. Data with a DOI should be further cited in the methods reference section.

REVIEWER COMMENTS:

Reviewer #1 (Remarks to the Author):

Please see attached review.

Reviewer #2 (Remarks to the Author):

Quantity and fate of eroded organic matter along the Arctic coastline is one of the least understood components of the Arctic carbon cycle. Improving our understanding of this challenging problem is therefore a high priority research area. The manuscript includes a lot of critical information on the distribution of organic carbon along coastal gradients, including near shore samples, demonstrating the disappearance of a certain, carbon-rich fraction between the nearshore and the "deeper" shelf areas. The team argues that this could indicate an underappreciated carbon loss in these systems that needs to be accounted for. I think this is an important study that contributes new and crucial information for understanding the role of climate-change-driven coastal erosion in the Arctic region.

I had one major concern about the current manuscript. What is the role of landfast ice for the removal of the low density, plant-material-rich organic matter along the coast. Could it be that a portion of the near shore eroded material is entrained in landfast ice and is subsequently removed from the near shore zone by ice export? What is the relative importance of physical removal by ice (no CO₂ source) versus the biological degradation of that organic carbon (CO₂ source). If you consider the timing of processes on the Arctic coast you realize that coastal erosion is highest during the fall season when it is warmest and the wave action is the highest along the coast. This results in maximum OC input from coastal erosion just before the freezing season. Much of this eroded soil OC will be incorporated into landfast ice and transported out of the area before the sediment is released back into the ocean. I think this issue needs to be addressed in the manuscript, including data from landfast ice in this region.

Communications Earth & Environment is committed to improving transparency in authorship. As part of our efforts in this direction, we are now requesting that all authors identified as 'corresponding author' create and link their Open Researcher and Contributor Identifier (ORCID) with their account on the Manuscript Tracking System prior to acceptance. ORCID helps the scientific community achieve unambiguous attribution of all scholarly contributions. You can create and link your ORCID from the home page of the Manuscript Tracking System by clicking on 'Modify my Springer Nature account' and following the instructions in the link below. Please also inform all co-authors that they can add their ORCIDs to their accounts and that they must do so prior to acceptance.

Version 1:

Decision Letter:

Dear Ms van Crimpen,

Your manuscript titled "Bridging the gap: shallow coastal zones are key mediators in Arctic land-ocean carbon fluxes" has now been seen by our reviewers, whose comments appear below. In light of their advice we are delighted to say that we are happy, in principle, to publish a suitably revised version in Communications Earth & Environment.

We therefore invite you to revise your paper one last time to address the remaining concerns of our reviewers, which pertain to discussing hydrodynamic sorting. At the same time we ask that you edit your manuscript to comply with our format requirements and to maximise the accessibility and therefore the impact of your work.

EDITORIAL REQUESTS:

*****Please take care to match our formatting and policy requirements. We will check revised manuscript and return manuscripts that do not comply. Such requests will lead to delays. *****

SUBMISSION INFORMATION:

OPEN ACCESS:

Communications Earth & Environment is a fully open access journal. Articles are made freely accessible on publication. For further information about article processing charges, open access funding, and advice and support from Nature Portfolio, please visit <https://www.nature.com/commsenv/open-access>

Link Redacted

Best regards,

Alice Drinkwater, PhD
Associate Editor
Communications Earth & Environment
Consulting Editor
Communications Sustainability

REVIEWERS' COMMENTS:

Reviewer #2 (Remarks to the Author):

I appreciate the addition of landfast ice export as an additional mechanism explaining the absence of LD, high C material in the outer shelf locations. I also appreciate the addition of references to this effect, even though most of the new references are now more than 20 years old and coastal erosion has increased substantially since then. One other aspect I wanted to bring up here for discussion is the role of the coastal current that clearly separates the near-shore from the far-shore sampling sites (see Figure 1 a). Is it possible that this current effects the hydrodynamic sorting of the different fractions differently, sweeping the LD, high-C sediment fraction preferentially towards the Canadian Archipelago, along the coast? Similar patterns have been observed with other riverine material which seems to reach the Beaufort Sea only ever fourth year or so.

Minor comment

Line 97: add space between sentences

** Visit Nature Portfolio's author and referees' website at www.nature.com/authors for information about policies, services and author benefits**

I read the manuscript by van Crimpen and colleagues with interest. They have focused on an under-studied part of the world – shallow Arctic coasts that receive erosion from permafrost – and considered the effect of carbon transport, degradation and burial in the 0-5 metre depths. The shelves in this region are extensive and shallow, meaning that the 0-5 metre zone extends far offshore, but is inaccessible by traditional research vessels.

They have measured the organic carbon content in various size and density fractions and related these to the hydrodynamic processes occurring in the nearshore zone. Their findings, including that plant material is prevalent nearshore but largely absent offshore, are important for those studying these Arctic coastlines. Large losses of organic matter in a nearshore 'biogeochemical reactor' could be an overlooked part of the carbon cycle.

I was impressed by the work and have no suggestions for improvement. I recommend publication in *Communication Earth and Environment*.

Response to reviewers for paper #COMMSENV-25-1185-T “Bridging the gap: shallow coastal zones are key mediators in Arctic land-ocean carbon fluxes”

Author responses to two anonymous reviewer comments are provided below. The reviewer comments are in *italics* followed by the responses in normal font. The line numbers in the responses refer to the **track-changed** document. Additions to the manuscript are highlighted in blue.

Reviewer #1:

I read the manuscript by van Crimpen and colleagues with interest. They have focused on an under-studied part of the world – shallow Arctic coasts that receive erosion from permafrost – and considered the effect of carbon transport, degradation and burial in the 0-5 metre depths. The shelves in this region are extensive and shallow, meaning that the 0-5 metre zone extends far offshore, but is inaccessible by traditional research vessels.

They have measured the organic carbon content in various size and density fractions and related these to the hydrodynamic processes occurring in the nearshore zone. Their findings, including that plant material is prevalent nearshore but largely absent offshore, are important for those studying these Arctic coastlines. Large losses of organic matter in a nearshore ‘biogeochemical reactor’ could be an overlooked part of the carbon cycle.

*I was impressed by the work and have no suggestions for improvement. I recommend publication in *Communication Earth and Environment*.*

Thank you for your positive comments and for your time reviewing this manuscript.

Reviewer #2:

Quantity and fate of eroded organic matter along the Arctic coastline is one of the least understood components of the Arctic carbon cycle. Improving our understanding of this challenging problem is therefore a high priority research area. The manuscript includes a lot of critical information on the distribution of organic carbon along coastal gradients, including near shore samples, demonstrating the disappearance of a certain, carbon-rich fraction between the nearshore and the “deeper” shelf areas. The team argues that this could indicate an underappreciated carbon loss in these systems that needs to be accounted for. I think this is an important study that contributes new and crucial information for understanding the role of climate-change-driven coastal erosion in the Arctic region.

I had one major concern about the current manuscript. What is the role of landfast ice for the removal of the low density, plant-material-rich organic matter along the coast. Could it be that a portion of the near shore eroded material is entrained in landfast ice and is subsequently removed from the near shore zone by ice export? What is the relative importance of physical removal by ice (no CO₂ source) versus the biological degradation of that organic carbon (CO₂ source). If you consider the timing of processes on the Arctic coast you realize that coastal erosion is highest during the fall season when it is warmest, and the wave action is the highest along the coast. This results in maximum OC input from coastal erosion just before the freezing season. Much of this eroded soil OC will be incorporated into landfast ice and transported out of the area before the sediment is released back into the ocean. I think this issue needs to be addressed in the manuscript, including data from landfast ice in this region.

We thank the reviewer for this insightful and important comment. We have now included a discussion of the role of landfast ice and sea ice export in the potential removal of low-density, plant-rich organic matter from the nearshore zone.

Specifically, we highlight that coastal erosion in the Arctic typically peaks in the late summer and fall, when wave action and temperatures are highest. This period of elevated input of OC from thawing permafrost soils and eroding tundra occurs just before seasonal freeze-up, raising the possibility that a portion of this OC becomes entrained in newly forming landfast or drifting sea ice.

We have added the following points to the discussion;
line 226-239

“Additionally, incorporation of sediments into ice during freeze-up in early fall may transport a portion of it offshore. This may be particularly relevant for sediment released via coastal erosion as this process is at its peak during late summer and early fall, at the same time when ice formation starts. On the Chukchi and Beaufort shelves, it is currently estimated that 5–8 Tg of eroded sediments become incorporated into the sea ice annually (Eicken et al., 2005), representing approximately 6.5-10% of the total 78 Tg sediment released annually by coastal erosion in that same area (Stein & Macdonald, 2004; in Wegner et al., 2015). These sediments, and their OC, can then be transported offshore and released during melting, potentially far from the source (Eicken et al., 1997; Foreman et al., 2000; Nürnberg et al., 1994). While a few studies suggest most of this sediment is fine-grained (Barnes et al., 1982; Eicken et al., 1997; Reimnitz et al., 1993), we cannot rule out the inclusion of OC-rich vascular plant fragments that are dominant in our study. We therefore suggest that the stark

decrease of the OC-rich LD fraction is likely a combination of mechanical and biological breakdown combined with transport processes offshore.

Added references on lines:

Line 380-382

Barnes P.W., Reimnitz E, & Fox D. (1982). Ice Rafting of Fine-Grained Sediment, a Sorting and Transport Mechanism, Beaufort Sea, Alaska. SEPM Journal of Sedimentary Research, Vol. 52. <https://doi.org/10.1306/212F7F86-2B24-11D7-8648000102C1865D>

Line 418-421

Eicken, H., Gradinger, R., Gaylord, A., Mahoney, A., Rigor, I., & Melling, H. (2005). Sediment transport by sea ice in the Chukchi and Beaufort Seas: Increasing importance due to changing ice conditions? Deep Sea Research Part II: Topical Studies in Oceanography, 52(24–26), 3281–3302. <https://doi.org/10.1016/j.dsr2.2005.10.006>

Line 422-424

Eicken, H., Reimnitz, E., Alexandrov, V., Martin, T., Kassens, H., & Viehoff, T. (1997). Sea-ice processes in the Laptev Sea and their importance for sediment export. Continental Shelf Research, 17(2), 205–233. [https://doi.org/10.1016/S0278-4343\(96\)00024-6](https://doi.org/10.1016/S0278-4343(96)00024-6)

Line 429-431

Foreman, M. G. G., Thomson, R. E., & Smith, C. L. (2000). Seasonal current simulations for the western continental margin of Vancouver Island. Journal of Geophysical Research: Oceans, 105(C8), 19665–19698. <https://doi.org/10.1029/2000JC900070>

Line 489-491

Nürnberg, D., Wollenburg, I., Dethleff, D., Eicken, H., Kassens, H., Letzig, T., Reimnitz, E., & Thiede, J. (1994). Sediments in Arctic sea ice: Implications for entrainment, transport and release. Marine Geology, 119(3–4), 185–214. [https://doi.org/10.1016/0025-3227\(94\)90181-3](https://doi.org/10.1016/0025-3227(94)90181-3)

Line 492-494

Parkinson, C. L., Cavalieri, D. J., Gloersen, P., Zwally, H. J., & Comiso, J. C. (1999). Arctic sea ice extents, areas, and trends, 1978–1996. Journal of Geophysical Research: Oceans, 104(C9), 20837–20856. <https://doi.org/10.1029/1999JC900082>

Line 500-502

Reimnitz, E., McCormick, M., McDougall, K., & Brouwers, E. (1993). Sediment Export by Ice Rafting from a Coastal Polynya, Arctic Alaska, U.S.A. Arctic and Alpine Research, 25(2), 83. <https://doi.org/10.2307/1551544>

Line 530-532

*Stein, R., & Macdonald, R. W. (2004). Geochemical proxies used for organic carbon source identification in Arctic Ocean sediments. In *The Organic Carbon Cycle in the Arctic Ocean* (pp. 24–32). Springer.*

Line 572-576

Wegner, C., Bennett, K. E., de Vernal, A., Forwick, M., Fritz, M., Heikkilä, M., Łącka, M., Lantuit, H., Laska, M., Moskalik, M., O'Regan, M., Pawłowska, J., Promińska, A., Rachold, V., Vonk, J. E., & Werner, K. (2015). Variability in transport of terrigenous material on the shelves and the deep Arctic Ocean during the Holocene. In *Polar Research* (Vol. 34, Issue 1). Taylor and Francis Ltd. <https://doi.org/10.3402/polar.v34.24964>

Response to reviewers for paper #COMMSENV-25-1185-T “Shallow coastal zones are key mediators in Arctic land-ocean carbon fluxes”

Author responses to two anonymous reviewer comments are provided below. The reviewer comments are in *italics* followed by the responses in normal font. The line numbers in the responses refer to the **track-changed** document. Additions to the manuscript are highlighted in blue.

Reviewer #2:

I appreciate the addition of landfast ice export as an additional mechanism explaining the absence of LD, high C material in the outer shelf locations. I also appreciate the addition of references to this effect, even though most of the new references are now more than 20 years old and coastal erosion has increased substantially since then.

One other aspect I wanted to bring up here for discussion is the role of the coastal current that clearly separates the near-shore from the far-shore sampling sites (see Figure 1 a). Is it possible that this current effects the hydrodynamic sorting of the different fractions differently, sweeping the LD, high-C sediment fraction preferentially towards the Canadian Archipelago, along the coast? Similar patterns have been observed with other riverine material which seems to reach the Beaufort Sea only ever fourth year or so.

We thank Reviewer #2 for this insightful suggestion regarding the role of the coastal current in hydrodynamic sorting of sediment fractions. We agree that this mechanism may further contribute to the observed spatial separation between nearshore and offshore sites, and we have added a reference to recent work supporting this process (Eidam et al., 2025).

Eidam, E. F., Stark, N., Nienhuis, J. H., Keogh, M. & Obelcz, J. Arctic Continental-Shelf Sediment Dynamics. *Ann Rev Mar Sci* 17, 435–460 (2025).

Line 97: add space between sentences

Adjusted

Added reference on line 473-474; Eidam, E. F., Stark, N., Nienhuis, J. H., Keogh, M. & Obelcz, J. Arctic Continental-Shelf Sediment Dynamics. *Ann Rev Mar Sci* 17, 435–460 (2025).